# Seized Ecstasy Pills: Infrared Spectra and Image Datasets

**Luc Patiny [1] [ID], Michaël Zasso [1], Pierre Esseiva [2] [ID] and Julien Wist [3,*] [ID]**

[1]  Zakodium Sàrl, 1027 Lonay, Switzerland; luc.patiny@epfl.ch (L.P.); michael@zakodium.com (M.Z.)
[2]  Institut de Police Scientifique, Université de Lausanne, 1015 Lausanne, Switzerland; Pierre.Esseiva@unil.ch
[3]  Chemistry Department, Universidad del Valle, Cali 76008, Colombia
*  Correspondence: julien.wist@correounivalle.edu.co

**Abstract:** According to the World Drug Report 2020, cocaine and ecstasy are the most consumed stimulant drugs, with 19 and 27 million estimated users in 2018. In this context, large efforts are being made to design fast and cost-effective analytical methods to track and monitor the distribution networks of these synthetic drugs. Here, we share two datasets of ecstasy pills seized in the northeast of Switzerland between 2010 and 2011. The first contains 621 forensic-grade images of pills, while the second one consists of 486 mid-infrared (mIR) spectra. While both sets are not covering the same seizure, both provide high-quality data with orthogonal information to evaluate clustering and dimension reduction methods.

**Keywords:** ecstasy; XTC; dataset; mIR; image analysis; machine learning

---

## 1. Summary

Amphetamine-type stimulants (ATS), mostly known as ecstasy, XTC, or designer drugs, are derivatives from Amphetamine (Am) and sold as tablets. The other most commonly found derivatives are MDA (3,4-methylenedioxyamphetamine) and MDMA (3,4-methylenedioxy-*N*-methylamphetamine) [1]. During 2018 only, 228 metric tons of methamphetamine (MAm) were seized globally [2]. The testing and monitoring of illicit pills is thus a gigantic task and a matter of public health. While many analytical platforms have been showcased for that purpose, the main focus has been the identification and/or quantification of active compounds in pills [3–6].

Instead, highlighting similarities between seizures as quickly as possible enables tracking the origin of the pills, hence unveiling trafficking routes and supply chains for production [7,8]. The size of this task calls for cheap methods with fast response time that can be deployed at a large scale as close as possible to where seizures occur. Such profiling methods exist that can either target the visual aspect of the pills, imaging [9], or their composition, optical spectroscopy [10,11], or even portable chromatography [12].

Both images and spectroscopy provide high-dimensional data, where a vast amount of variables, pixels, or frequencies are measured simultaneously. Consequently, this data is usually large, and it requires of mathematical models to extract insightful information, such as most relevant variables, or to use it as classifiers. Although high-dimensional data can always be represented as $m \times n$ matrices, data processing depends on the nature of the data, i.e., spectral resolution, number of points, quality of the baseline, and the signal-to-noise ratio, among others. Images will be analyzed by first

defining regions of interest and then applying filters, for example. Then, different multivariate analysis strategies could be tested to extract information. Non-supervised approaches, such as Principal Component Analysis, Non-negative Matrix Factorization [13], or more recently, Uniform Manifold Approximation and Projection (UMAP) [14] are most commonly used. Again, the choice of the method is dictated by the type of data, the experimental design, and the aim of the research. It is not unusual to benchmark different data processing and normalization techniques, and analysis pipelines to pick the most appropriate one. On the other hand, the development of new statistical models requires access to high-quality and diverse datasets to evaluate their performance.

The two datasets described herein provide orthogonal information, visual aspects (images), and chemical composition (mIR spectra) about different seizures of ecstasy pills seized in the streets of Switzerland. The large amount of data made it challenging to cluster the seizures in a single analysis.

## 2. Data Description

The first dataset consisted of 602 images (4310 × 2868 pixels) in png format, and was used in previous work by the authors of this manuscript [9], but not published on that occasion. The size of each file was approximately 1 Mb and named after the seizure and a sequential pill number in that seizure. For example, the file 1121_2.png contained a picture of pill 2 of seizure 1121. There were 145 different seizures, and for each seizure, 2 to 19 pills have been pictured, as summarized in Table 1.

**Table 1.** Number of pills pictured for each seizure. A total of 602 pills were pictured that represented 145 different seizures.

| Number of Seizures | 19 | 1 | 99 | 2 | 2 | 1 | 2 | 1 | 5 | 13 |
|---|---|---|---|---|---|---|---|---|---|---|
| Number of Pills Analyzed | 10 | 19 | 2 | 3 | 4 | 5 | 6 | 7 | 8 | 9 |

The second dataset contains 486 mid-infrared (mIR) spectra (650 to 4000 $cm^{-1}$) of 6701 points that were acquired at the same period, 2011, but never published. They represent 41 different seizures, and several replicates were analyzed for each of them, as described in Table 2.

**Table 2.** Number of pills analyzed for each seizure. A total of 486 pills were analyzed that represent 41 distinct seizures.

| Number of Seizures | 8 | 2 | 19 | 1 | 9 | 2 |
|---|---|---|---|---|---|---|
| Number of Pills Analyzed | 10 | 14 | 15 | 21 | 6 | 6 |

The files are in JCAMP-DX [15] format and are named after each seizure lot. For example, the file 0327_10.png contains the spectrum of pill number 10 of seizure lot 0327. For each seizure, replicated measurements were performed. Either several pills measured from the same seizure were measured independently, or several spectra were recorded of the same pill. Both types of replicates were labelled differently. For instance, the file "mdmaHCL00420_1" contains the spectra of the pill number 1 of seizure "0420", while "1693_1_a" indicates spectra "a" of pill 1 or seizure "1693". Thus, the trailing number indicates the number of the pill, while the letter identifies replicated measurement of the same pill. This information was also available in the `TITLE` section of the .jdx files.

Since reading JCAMP files may be discouraging for non-chemists, two files in tsv format are provided, `matrix.tsv` and `matrix\_snv.tsv`. The former is the original data without processing, while the Standard Normal Variate (SNV) transformation [16,17] was applied to create the latter. The first five columns describe the ID, title, color, display, and category of the spectra stored in each row, and are required by our tools for analysis and visualization [18]. The remaining columns contain the absorbance values, while the header of each column describes the wavenumber [$cm^{-1}$]. This tsv format is readily read by most software, including MS Excel, R, python, and so forth.

Although the dataset covers different seizures, the data cannot be used in direct comparison. However, it is worth mentioning that 31 seizures are common to both datasets.

## 3. Methods

### 3.1. Imaging

All images were acquired by trained forensic staff at Université de Lausanne using well-established protocols and standard equipment, as depicted in Figure 1. While this process was useful to evaluate the prospects of such an approach, a future implementation must perform well with data acquired from different sources, including hand-held smartphones.

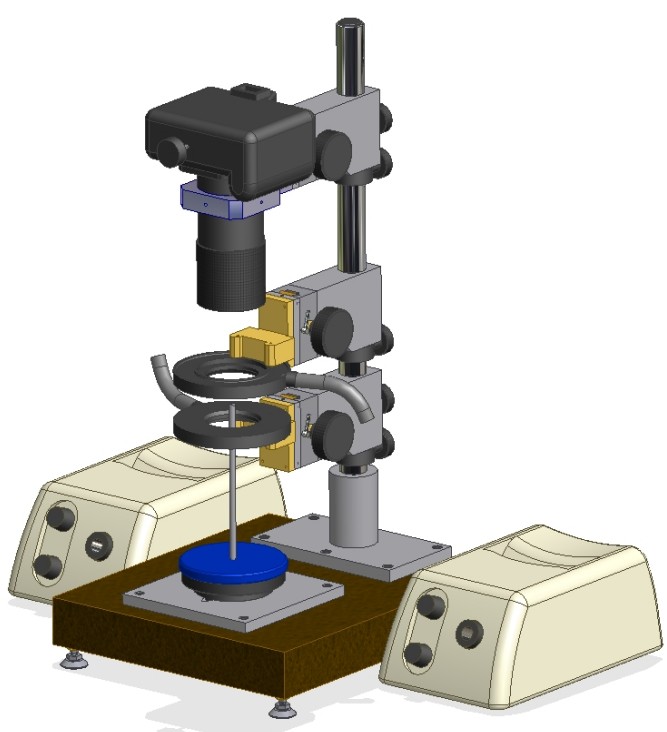

**Figure 1.** The schema of the experimental setup for depicting each pill. This setup is standard to forensic proceedings. The camera is a Canon D90 with Canon EFS 60 mm lens. The two black rings ensure homogenous lightning from two independent light sources (Marcel Aubert SA, MA 1300, Nidau, Switzerland). The pill is placed on the vertical rod in the center and a white square of paper is used as a reference for color and size.

### 3.2. mIR Spectra

All spectra were acquired using a Nicolet iS5 spectrophotometer (Thermo Fisher Scientific, Waltham, MA, USA) equipped with a iD5ATR module and using the parameters as described in Table 3.

**Table 3.** Experimental parameters to record the mIR spectra.

| | |
|---|---|
| Crystal | Diamond, single reflexion (45) $n$ = 2.419 |
| Range | 4000–650 cm$^{-1}$ |
| Number of Scans | 32 |
| Resolution | 4 cm$^{-1}$ |
| Collection Length | 52 s |

Each sample was ground to a fine powder, homogenized, placed, and maintained over the crystal with the same pressure. Prior to each measurement, the crystal was carefully washed and a blank spectrum acquired.

## 4. Usage Note

To the extent of our knowledge, both datasets are the first of their kind to be published, probably due to the access restrictions that apply to police evidence material and, in this case, to controlled substances. The release of this original dataset will allow researchers from different fields to evaluate and propose new suitable strategies for extracting information from such data, and thus will contribute to finding solutions to a very acute public health issue.

Infrared spectra shows that most of the pills in this dataset contain 3,4-Methylenedioxy methamphetamine (MDMA) as an active compound, which is expected according to recent reports [2]. However, some pills contain different derivatives, as indicated by the large deviations observed in their spectra (See Figure 2). Other molecules are necessary for the preparation of the tablets. These latter can also be used to profile seizures that may share the same supply chain or prepared following a similar recipe. The consistency of the composition found among different pills from the same seizure, such as when replicates are available, provides additional information about the production method. Images, on the other hand, provide a rapid means of discriminating different distribution networks, although different-looking tablets may be produced by the same laboratory as an effort to confuse police investigations.

Smartphones are already universally available. They enable users to upload geolocalized images of the seized pills and retrieve information about previous seizures that contain similar tablets. An example of clustering analysis for that purpose was demonstrated several years ago by the authors, and readers are referred there for further details [9]. Hence, this dataset could prove useful to gauge the impact of recent progress in machine learning and artificial intelligence, in terms of speed and accuracy. These data could be used to develop smartphone applications that enable users to quickly check if the product they purchased has been reported as dangerous.

Portable and ultra-portable spectrophotometers are becoming available and may be used during routine police controls or raids to complement visual information. However, other applications could be envisioned to enable real-time assessment of the quality or toxicity of the tablets and reporting it to a central database.

An example of a simple multivariate analysis, PCA, is shown below using online and open source tools developed for the electronic notebooks c6h6.org. Figure 2 shows an overlay of the 486 spectra after applying a standard normal variate transformation and is color-coded by seizure. The upper portion shows a score plot of the first two principal components. To access this tool, it is preferable to use Google Chrome, and while on the landing page, one should choose the "PCA" tile. Enter "XTC" in the search bar on the left to retrieve the data. Seizures can be selected individually using the "+" buttons, or added as a whole using the "+" sign located on the header of the "List of selected sample" window. To calculate the PCA or to re-calculate PCA after removing outliers (using alt-Draw in the score plot), use the button "calculate/recalculate PCA".

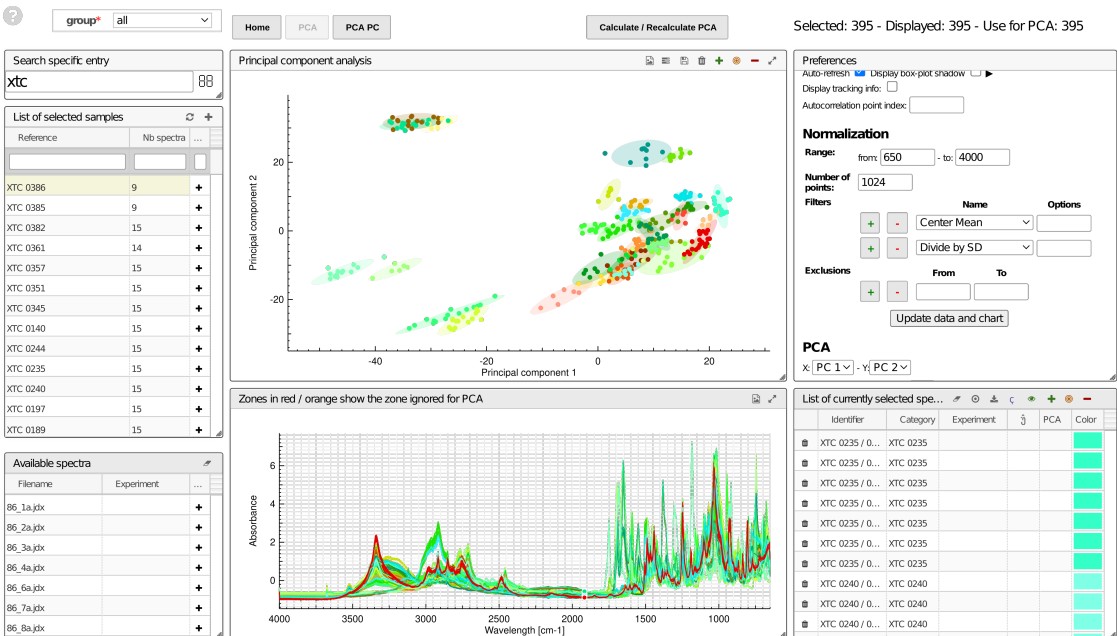

**Figure 2.** Screenshot of c6h6.org electronic notebook showing a principal component analysis of 486 spectra of ecstasy. The bottom window depicts the spectra, while the upper one shows the resulting score plot for the first two principal components. The separation observed in the direction of the first principle component is mainly accounted for by the spectral region from 1500 to 1750 according to loading plots (not shown) and is presumed to reflect the presence of different compounds.

As expected, some seizures are clustered very compactly (seizure 1132, 0966 and 0244, uppermost cluster), while some clusters appear very distant (seizure 0140, leftmost cyan cluster). Visual inspection of the pills from seizures, 1132, 0244, and 0966 suggests all three originate from the same source, while the pill from seizure 0140 clearly looks different (See Figure 3).

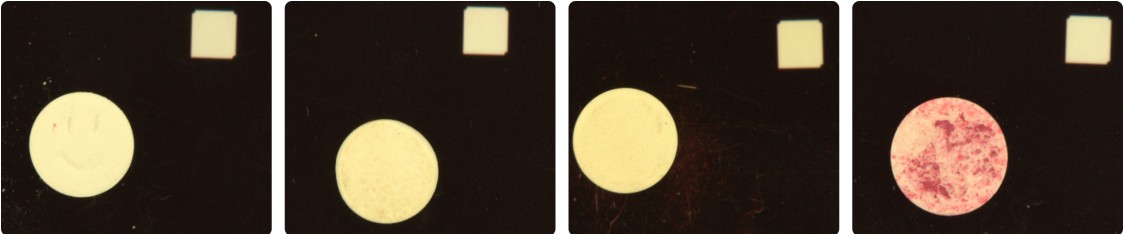

**Figure 3.** From left to right, images of pills from seizure 0244, 1132, 0966, and 0140.

Hierarchical clustering analysis is another widely used approach to explore structure in data. The resulting clustering, shown in Figure 4, was achieved using the spectra similarity tool available in the open source c6h6 notebook. In this case, only the four seizures 0244, 1132, 0966, and 0140 were used as inputs. As expected, the former were clustered together as they looked similar in shape and colour, while the latter group was singled out because of its colour. (See Figure 4). Seizures 0244, 1132, and 0966 could not be further classified, as observed on the close-up on the right, and were thus presumed to originate from the same source.

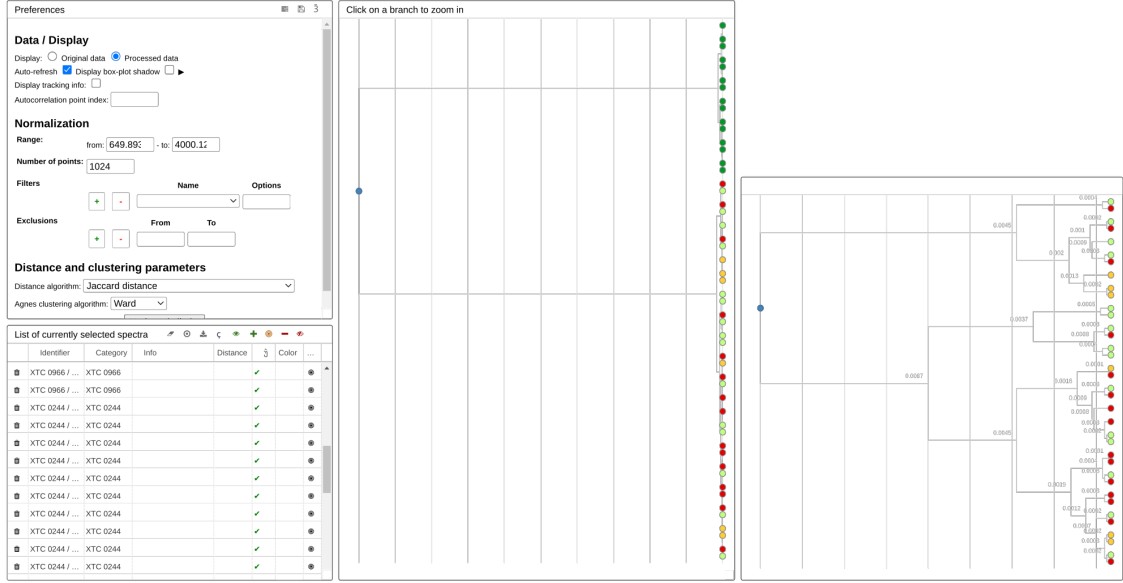

**Figure 4.** Clustering of four seizures, 1132, 0244, 0966, and 0140. The middle panel shows the full tree. The upper branch groups all the pills from seizure 0140, while the lower branch aggregates all the remaining pills that were likely issued by the same laboratory. The panel on the right zooms in on the lower branch. As expected, no classification was observed for that branch, confirming that the pills were from a similar source.

**Author Contributions:** Conceptualization, J.W. and L.P.; Data curation, L.P. and M.Z.; Project funding and supervision, P.E.; Analysis and visualization packages, M.Z.; Drafting the manuscript, J.W.; Revision and edition, L.P. and P.E. All authors have read and agreed to the published version of the manuscript.

**Funding:** This research received no external funding.

**Conflicts of Interest:** The authors declare no conflict of interest.

## Abbreviations

| | |
|---|---|
| MDPI | Multidisciplinary Digital Publishing Institute |
| DOAJ | Directory of Open Access Journals |
| TLA | Three Letter Acronym |
| LD | Linear Dichroism |

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
