# Peer review of "Seized Ecstasy Pills: Infrared Spectra and Image Datasets"

_data, 2010_

Round 1
Reviewer 1 Report
The attempt to locate the origin of the seizures is fundamental in the fight against drug trafficking. This work presents an important approach in this regard. It is therefore of great importance to be able to identify the different batches in the seizures and the possible similarities that can help to identify the origin.
Both the images and the spectra of the seized substances can be of great help. However, as the authors themselves point out, the producing laboratory could manufacture tablets of different appearance in an attempt to avoid tracking and tracing the production. In addition to the active molecules, other components are necessary for the preparation of the tablets. And these additional components can be used to identify products from the same batch, since they safely share the same excipients.
In any case, it would be necessary to try to apply this comparison of methods to the same samples (from the same seizures) so that the resulting information could be contrasted and compared. It is difficult to understand that samples seized in the years 2010 and 2011 are used. An attempt should have been made to do this study with samples of recent seizure in order to be able to update data and knowledge about the situation at the present time.
It would be necessary to advance in this knowledge so that the data provided by these tools can be obtained in a simpler way, within the reach of most laboratories. For this it would be necessary to implement a tool, as the authors suggest, more accessible. The development of an application for mobile phones would allow access to important information, even for the users themselves.
This work supposes a good approach but there is therefore still a long way to go in this type of research, to give it a real practical application.
Author Response
We thank the referee for the kind suggestions. Indeed there is a lot more work that has to be done before a practical application. This is one of the major motivations for us to publish these data, that are difficult to obtain so that others could test and validate their ideas. The seizures were performed from 2010 to 2011, and the analysis and publication was achieved in 2011. We believe that the data are still of interest, since during that period no other dataset was published on that topic.
At the time when the data were collected, smartphones were not as widely available as they are now. However in all software development it is generally accepted that high quality data are used in preliminary rounds, and later adapted to lower quality data. This data set represents this high quality set acquired under normalized conditions, while lower quality images can always be acquired in a later stage or created from the high quality images if preliminary results are convincing.
Reviewer 2 Report
Remarks:
a. In my opinion it is absolutely necessary to add, at the end of the manuscript, a discussion section.
b. Present screenshots (Fig. 2 and 4) need to be described and analyzed in more detail. Reasons for the observed clustering should be indicated.
c. Does the second data set contain 486 or 480 spectra?,
d. A number of mistakes can be found in the reference section.
Below I list some lines with statements containing errors or not clear to me:
3, 13-14, 24-25, 31-32, 42-44, 48-50, 52, 55-60, 89-98.
Author Response
- In my opinion it is absolutely necessary to add, at the end of the manuscript, a discussion section.
R: We agree with the referee that a discussion is important. Following the template provided by the journal, we discussed the results and the potential use of the dataset in the “usage note” section, since the template does not provide an explicit section for discussion. We would be happy to rename this section if it fulfils the journal’s guidelines.
- Present screenshots (Fig. 2 and 4) need to be described and analyzed in more detail. Reasons for the observed clustering should be indicated.
R: Both captions have been extended.
- Does the second data set contain 486 or 480 spectra?
R: The dataset contains 486 spectra
- A number of mistakes can be found in the reference section.
R: Indeed, this section has been revised and corrected.
Below I list some lines with statements containing errors or not clear to me:
R: 3 The sentence has been modified
R: 13-14 the sentence has been corrected
R: 24-25 the reference has been corrected
R: 31-32 the sentence has been corrected!
R: 42-44 sentence has been rewritten
R: 48-50 in order to avoid confusion in the manuscript we have replaced batches by seizures. The sentence has also been rewritten.
R: 52 We have added information about the fact that it is mid-infrared spectra.
R: 55-60 This paragraph has been entirely rewritten.
R: 89-98. This paragraph has been entirely rewritten.